# Strain-driven Kovacs-like memory effect in glasses

Yu Tong[1], Lijian Song[1,2] ✉, Yurong Gao[1], Longlong Fan[3], Fucheng Li [4], Yiming Yang[3], Guang Mo[3], Yanhui Liu [4], Xiaoxue Shui[5], Yan Zhang [1], Meng Gao[1], Juntao Huo [1,2], Jichao Qiao[6], Eloi Pineda [7] ✉ & Jun-Qiang Wang [1,2] ✉

Studying complex relaxation behaviors is of critical importance for understanding the nature of glasses. Here we report a Kovacs-like memory effect in glasses, manifested by non-monotonic stress relaxation during two-step high-to-low strains stimulations. During the stress relaxation process, if the strain jumps from a higher state to a lower state, the stress does not continue to decrease, but increases first and then decreases. The memory effect becomes stronger when the atomic motions become highly collective with a large activation energy, e.g. the strain in the first stage is larger, the temperature is higher, and the stimulation is longer. The physical origin of the stress memory effect is studied based on the relaxation kinetics and the in-situ synchrotron X-ray experiments. The stress memory effect is probably a universal phenomenon in different types of glasses.

The nature of glass is one of the most puzzling and challenging problems in condensed matter physics. One crucial aspect is the complex relaxation behaviors that arise from nonequilibrium thermodynamics. When glasses are subjected to external stimulations, e.g. stress or elevated temperatures, they usually relax monotonically towards lower energy states[1–4]. But some anomalous nonmonotonic relaxations may happen under multi-step annealing, which was observed by Kovacs in the 1960s[5]. Since then, the enthalpy/volume rejuvenation phenomenon after two-step annealing is also known as the Kovacs memory effect[5,6]. Such memory effect is revealed to be related to the heterogeneous nature of glasses[7,8], which can be described by phenomenological models such as the Tool-Narayanaswamy-Moynihan (TNM) model[2,6,9–14], and is closely related to the complex relaxation paths in the deep aging stage[15]. Thus, studying complex relaxation behaviors is of vital importance for understanding the non-equilibrium nature of glasses.

Besides the classical Kovacs memory effect in structural glasses that are triggered by the multi-step thermal annealing process[15–20], the Kovacs-like memory effect has been widely observed in other complex disordered systems[14,21–26]. Multistep mechanic stimulation can trigger the memory effect in the disordered granular system, crumpled thin sheets, and elastic foams[2,14,18,23,27,28]. Multistep thermal and magnetic stimulations can trigger the memory effect in disordered magnetic nanoparticles[21,29,30]. For structural glasses, recent results suggest that the stress and the thermal effects both contribute similarly to triggering flow[31–33]. However, the mechanic-field-triggered memory effect has been rarely observed in structural glasses[21]. It is intriguing to study whether structural glasses exhibit Kovacs-like memory effect under multiple strain stimulations.

In this Letter, we report a Kovacs-like memory effect in structural glasses, which is triggered by the two-step strain-stimulating procedure. This phenomenon is probably universal in glasses as it is

[1]CAS Key Laboratory of Magnetic Materials and Devices, and Zhejiang Province Key Laboratory of Magnetic Materials and Application Technology, Ningbo Institute of Materials Technology and Engineering, Chinese Academy of Sciences, Ningbo, China. [2]Center of Materials Science and Optoelectronics Engineering, University of Chinese Academy of Sciences, Beijing, China. [3]Beijing Synchrotron Radiation Facility, Institute of High Energy Physics, Chinese Academy of Sciences, Beijing, China. [4]Institute of Physics, Chinese Academy of Sciences, Beijing, China. [5]Ningbo Institute of Materials Technology and Engineering, Chinese Academy of Sciences, Ningbo, China. [6]School of Mechanics, Civil Engineering and Architecture, Northwestern Polytechnical University, Xi'an, China. [7]Department of Physics, Institute of Energy Technologies, Universitat Politècnica de Catalunya, Barcelona, Spain. ✉ e-mail: songlj@nimte.ac.cn; eloi.pineda@upc.edu; jqwang@nimte.ac.cn

observed in two metallic glasses (MGs) and a polymer glass (Polyvinyl chloride, PVC) with distinct properties. The influence of applied strains, temperatures, and preloading times have been systematically studied, and the relationship between the memory effect and relaxation kinetics is unveiled. In-situ high energy X-ray diffraction is applied to unravel the micro-structure origin of the memory effect.

## Results

Figure 1a illustrates the two-step tensile protocol, starting with an initial strain $\varepsilon_0 = 0$. Subsequently, the strain holds at a preloading strain of $\varepsilon_1$ for time $t_w$ and then jumps to another stress-relaxation strain of $\varepsilon_2$. Figure 1b shows the representative stress relaxation curves for $Ti_{16.7}Zr_{16.7}Hf_{16.7}Cu_{16.7}Ni_{16.7}Be_{16.7}$ high-entropy MG with $\varepsilon_1 = 0.5\%$ at a tested temperature $T_a = 593$ K. The dynamic characterization is presented in Supplementary Fig. 1. During the preloading step, the stress decays monotonically as expected. In the second step, when $\varepsilon_2 > \varepsilon_1$, the stress continues to decay monotonically. However, when $\varepsilon_2 < \varepsilon_1$, the stress increases first and then decreases after reaching a maximum. Such a non-monotonic change in stress is analogous to the classical Kovacs memory effect of enthalpy/volume induced by low- to-high two temperatures annealing[5,34]. For the same $\varepsilon_1$, a smaller $\varepsilon_2$ gives a stronger stress rejuvenation. The anomalous nonmonotonic evolution in stress, here named the stress memory effect, is also observed in two other glasses with distinct chemical compositions, $(Fe_{11}Zr_1)_{91.2}B_{8.8}$ MG and Polyvinyl chloride (PVC), as shown in Fig. 1c. The three representative glasses cover a wide range of physical properties, e.g., glass transition temperature, fragility, and yield strength, suggesting that such a stress memory effect is probably a universal characteristic of glasses.

The influence of preloading strain $\varepsilon_1$ and annealing temperature on the stress memory effect are summarized in Fig. 2. Figure 2a shows stress evolution in the second iso-strain stage for different pre-loading strains $\varepsilon_1$. The memory strength $\Delta\sigma$ and the peak time $t_p$ increase when the pre-loading strain increases, as shown in Fig. 2a–c. Figure 2d shows the stress memory effect in the second iso-strain stage at different temperatures $T_a$. The peak time $t_p$ increases with temperature (see Fig. 2e). The memory strength $\Delta\sigma$ also increases with temperature and increases faster at higher temperatures, as shown in Fig. 2f. The non-monotonic normalized stress can be well fitted by double Kohlrausch-Williams-Watts (KWW) expression[3,35,36]:

$$\frac{\sigma(t)}{\sigma_0} = A \cdot \exp\left(-\left(\frac{t'}{\tau_1}\right)^{\beta_1}\right) + (1-A) \cdot \exp\left(-\left(\frac{t'}{\tau_2}\right)^{\beta_2}\right) \quad (1)$$

where $\sigma(t)$ and $\sigma_0$ represent the immediate and initial stress in the second stage, respectively; $t' = t-t_w$ is the time in the second stage; $\tau_1$ and $\tau_2$ are the characteristic relaxation times; $\beta_1$ and $\beta_2$ are the stretching exponents describing the non-exponential behavior of the relaxations. $A$ and $1-A$ reflect the corresponding relaxation intensity. The fitted relaxation time is shown in Supplementary Fig. 3.

The influence of the pre-loading time $t_w$ on the stress memory effect is studied for three representative glasses, as shown in Fig. 3a–c. The dependence of the memory strength $\Delta\sigma$ on pre-loading time is shown in Fig. 3d–f. Generally, the $\Delta\sigma$ increases monotonically as the preloading time increases. It is interesting to note that there exists a critical time, before which the $\Delta\sigma$ is very small, but after which the $\Delta\sigma$ increases sharply. This is a notable result of the stress memory effect, which has not been reported in disordered granular systems.

## Discussion

To study the physical origin of the emerging stress memory effect, the relaxation activation energy $E_a$ during the preloading stage was calculated according to the KWW and Arrhenius equations[3,37,38] (see details in section 4 of Supplementary information). The time-dependent $E_a$ for the three representative glasses during the preloading step are shown in the upper panels of Fig. 4a–c. Considering Figs. 3 and 4, the $E_a$ is small and does not change much with relaxation time while the stress memory effect is weak. When the relaxation time is long enough where the stress memory becomes strong, the $E_a$ increases fast. The analogous change in $E_a$ during isothermal annealing was reported in different glasses[35,39–41], and it is attributed to the increased atomic collectivity during relaxation.

Furthermore, we studied the strain recovery behavior after the preloading, see Supplementary Fig. 7. Below the glass transition temperature $T_g$, the mechanical response of the glassy system usually includes the instantaneous recoverable elastic strain $\varepsilon_{el}$, slow recoverable anelastic strain $\varepsilon_{an}$, and irrecoverable viscoplastic strain $\varepsilon_{vp}$, as distinguished in Supplementary Fig. 7a. The fraction of strain recovery $(1 - \varepsilon_{vp}/\varepsilon_1)$ of three glasses after various preloading times $t_w$ are shown in the lower panels of Fig. 4. When the $t_w$ is short, the glasses exhibit complete strain recovery. When the $t_w$ is long enough where stress memory thrives, the glass cannot recover completely, which suggests irreversible configuration during the preloading process[42]. The reversible process is clearly associated with the small $E_a$ (see the upper panels of Fig. 4), and it should be related to the activation and motion of localized soft domains that are constrained by the elastic matrix[40–44].

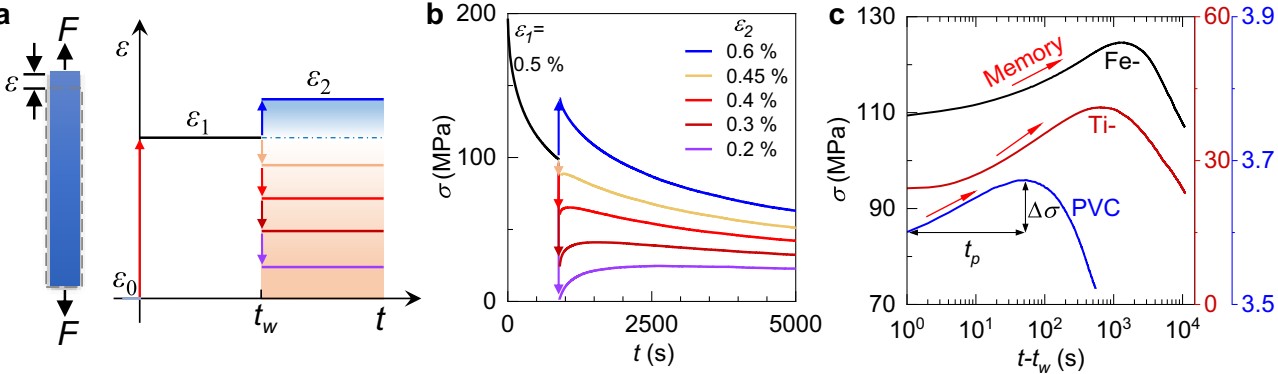

**Fig. 1 | Two-step strain annealing protocol and stress memory effect. a** The schematic of tensile stress relaxation with two different applied strains ($\varepsilon_1$, $\varepsilon_2$) with an initial strain $\varepsilon_0 = 0$. The preloading time is $t_w$. $F$ represents axial tensile force. The blue gradient represents increasing strain, while the red gradient represents decreasing strain. **b** The stress relaxation curves of two-step strain stimulations ($\varepsilon_1 = 0.5\%$, $\varepsilon_2 = 0.2\% \sim 0.6\%$) for TiZrHfCuNiBe metallic glass at $T_a = 593$ K. **c** The semi-logarithmic plots of non-monotonic stress relaxation curves for three glasses

(Red: TiZrHfCuNiBe high-entropy MG with the preload parameters of ($\varepsilon_1 = 0.5\%$, $\varepsilon_2 = 0.3\%$, $t_w = 900$ s, $T_a = 593$ K); Black: $(Fe_{11}Zr_1)_{91.2}B_{8.8}$ MG with the preload parameters of ($\varepsilon_1 = 0.4\%$, $\varepsilon_2 = 0.2\%$, $t_w = 1800$ s, $T_a = 597$ K); Blue: PVC with the preload parameters of ($\varepsilon_1 = 0.5\%$, $\varepsilon_2 = 0.3\%$, $t_w = 300$ s, $T_a = 318$ K)). The titles of all three y-axis are $\sigma$ (MPa). The $t_p$ and $\Delta\sigma$ denote the peak time and the strength of the memory effect, respectively. Source data are provided as a Source Data file.

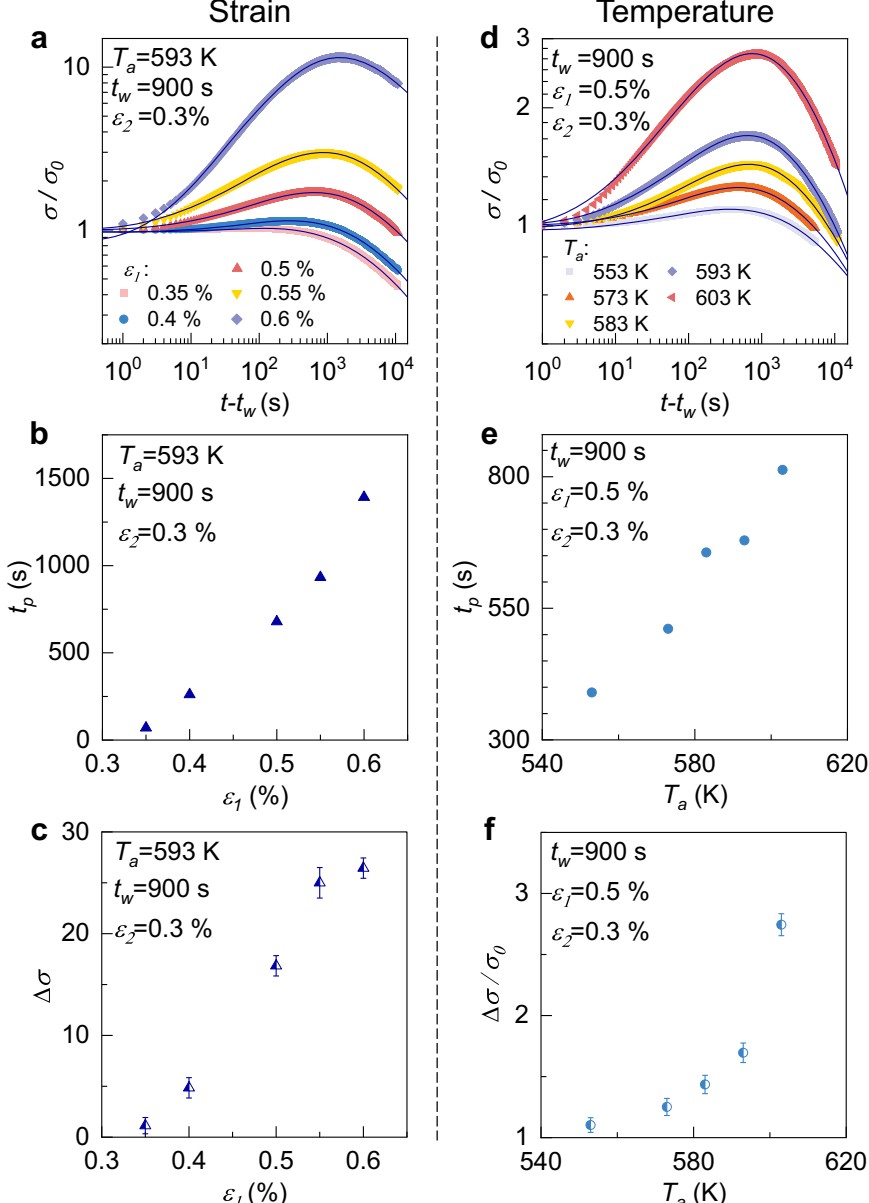

**Fig. 2 | Influences of the preloading strain and temperature on stress memory effect for TiZrHfCuNiBe high-entropy MG. a** The stress memory effect for different preloading strains $\varepsilon_1$. The dependence of **b** peak time $t_p$ and **c** the strength of memory effect $\triangle\sigma$ on $\varepsilon_1$. **d** The stress memory effect at different annealing temperatures $T_a$. The dependence of **e** peak time $t_p$ and **f** the relative strength of memory effect $\triangle\sigma/\sigma_0$ on $T_a$. The solid curves correspond to double KWW fitting. Error bars represent standard deviations. Source data are provided as a Source Data file.

As for the long-time annealing, the irrecoverable viscoplastic strain emerges associated with the sharp increase in $E_a$, and it derives from the percolation and cooperative motions of soft domains[42,45]. The fast and slow relaxation processes can be identified respectively as $\beta$ process and collective motion[39,40,42,46].

To unravel the structural origin of the stress memory effect, we utilize high energy synchrotron X-ray diffraction to investigate the atomic scale structural changes in TiZrHfCuNiBe MG during the stress memory effect, and the experimental technical details are displayed in Supplementary Fig. 8. Figure 5a, c shows the reduced pair distribution functions $G(r)$ of TiZrHfCuNiBe MG during the preloading stage and during the memory stage, respectively (stress evolution curve shown in Supplementary Fig. 8c). Close inspections indicate that the first nearest neighbor shell split into two sub-peaks (insets in Fig. 5a, c), in which the left sub-peak corresponds to the shorter inter-atomic band lengths dominated by small-size atoms, while the larger atoms

dominate the right sub-peak[46–48]. According to the interatomic bond lengths and weight factors provided in Supplementary Table 1, the left sub-peak at approximately 2.70 Å can be primarily attributed to the Hf-Cu and Hf-Ni atomic pairs, while the right sub-peak at around 3.10 Å is mainly dominated by the Zr-Hf, Hf-Hf, and Ti-Hf atomic pairs. As the stress decays, the full width at half maximum (FWHM) of the first $G(r)$ peak decreases (Fig. 5b), together with the left-shifted peak position, it suggests that the atomic packing becomes denser at a longer time[47]. But when stress increases during the memory stage (Fig. 5d), the FWHM increases and the peak position shifts to the right-hand side. This suggests that the atomic packing in the short-range becomes loose.

The anelastic recovery tests under the various preloading conditions in TiZrHfCuNiBe MG were performed to investigate the reversible contribution to the deformation, as illustrated in Supplementary Fig. 9. The anelastic recovery process can be fitted by the KWW

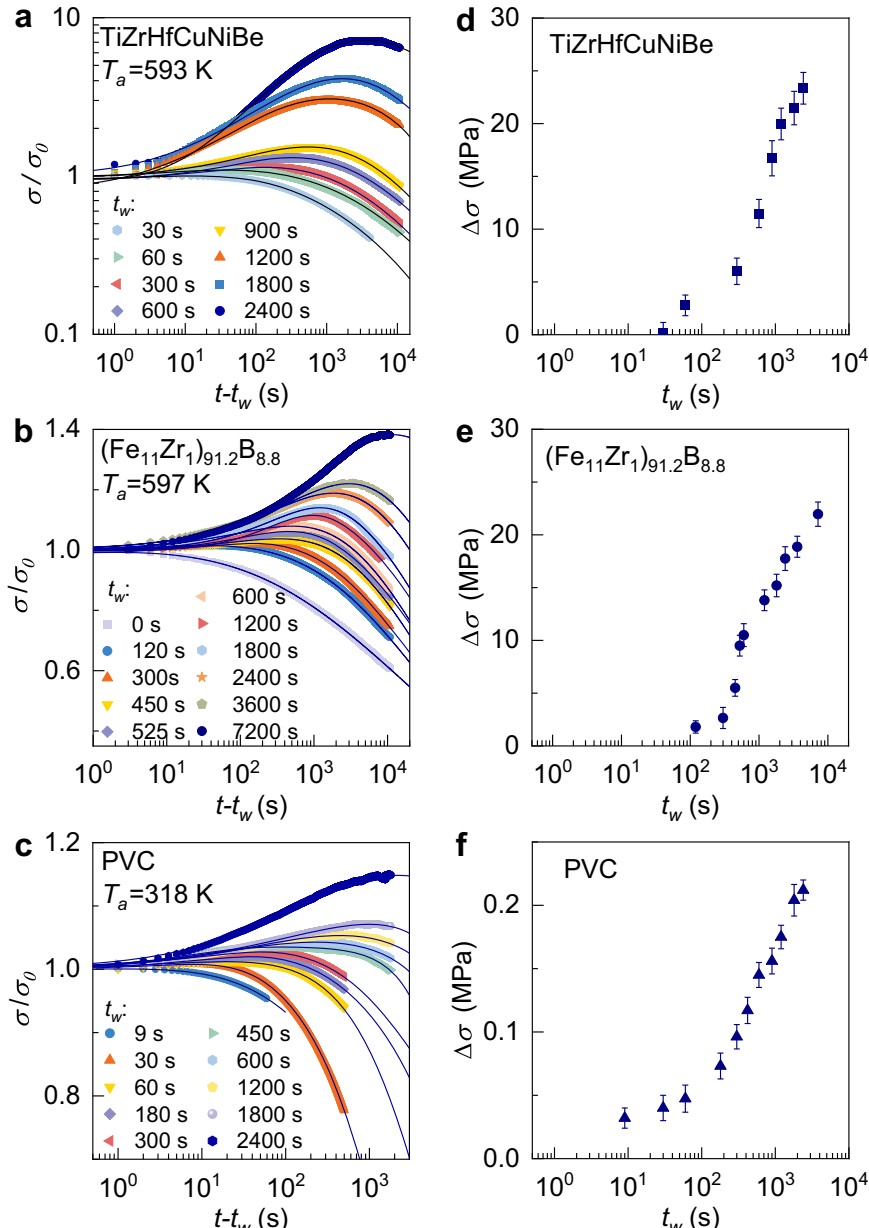

**Fig. 3 | Evolution of stress memory effect with annealing time for three glasses.** **a** TiZrHfCuNiBe high-entropy MG at 593 K with $\varepsilon_1 = 0.5\%$ and $\varepsilon_2 = 0.3\%$, **b** (Fe$_{11}$Zr$_1$)$_{91.2}$B$_{8.8}$ MG at 597 K with $\varepsilon_1 = 0.4\%$ and $\varepsilon_2 = 0.2\%$, and **c** PVC polymer glass at 318 K with $\varepsilon_1 = 0.5\%$ and $\varepsilon_2 = 0.3\%$. The solid curves correspond to KWW fitting.

The strength of the memory effect ($\Delta\sigma$) as a function of $t_w$ are shown in **d**–**f** accordingly. The blue shaded area highlights the fast increase of memory strength $\Delta\sigma$. Error bars represent standard deviations. Source data are provided as a Source Data file.

equation: $\frac{\varepsilon_{an}(t)-\varepsilon_{an}(\infty)}{\varepsilon_{an}(0)-\varepsilon_{an}(\infty)} = \exp(-(\frac{t}{\tau_{rec}})^{\beta_{rec}})$[37,39]. The relaxation characteristic time $\tau_{rec}$ reflects the timescale of the anelastic recovery[3,42,46,49]. The $\varepsilon_{an}(\infty)$ represents the ultimate strain achieved at the end of the anelastic recovery, and $\beta_{rec}$ is a stretching exponent. The fitting curves for various temperatures $T_a$, preloading strains $\varepsilon_1$ and times $t_w$ are shown in Supplementary Fig. 9. Figure 5e confirms a linear relationship between the peak time $t_p$ and the recovery characteristic time $\tau_{rec}$ over several decades. This indicates that the stress memory effect originates from the recovery of reversible anelastic components.

Figure 5f shows a schematic illustration of the local atomic arrangement for reversible events in stress relaxation. As the strain increases ($\varepsilon_1 \rightarrow \varepsilon_{II}$), the atoms pack more densely. This reconfiguration arises from the diffusion of small atoms within the network formed by the large solvent atomics, and it can be related to the left

shift of the first sub-peak (Fig. 5a). During the memory stage when unloading to a lower strain ($\varepsilon_{II} \rightarrow \varepsilon_1$), the small atoms move toward a looser configuration according to the right shift of the first sub-peaks (Fig. 5c). This observation coincides with the atomic mechanisms of the dynamic process in $\beta$ relaxation[47,50]. This result indicates that the stress memory effect arises from reversible $\beta$ relaxations.

Recent evidence has stated that the rejuvenation of glasses could be induced by the mismatch of the thermal expansion coefficient or the modulus fluctuation between neighboring local domains[51–53] when subjected to the same external perturbation. For the stress memory effect, as the applied strain jumps to a lower value, the mismatch of momentary modulus between neighboring local domains produces internal stresses, which would induce an inverse process and increase the stress. Longer preloading can stimulate abundant internal stress and facilitate more runback of reversible $\beta$ events upon unloading,

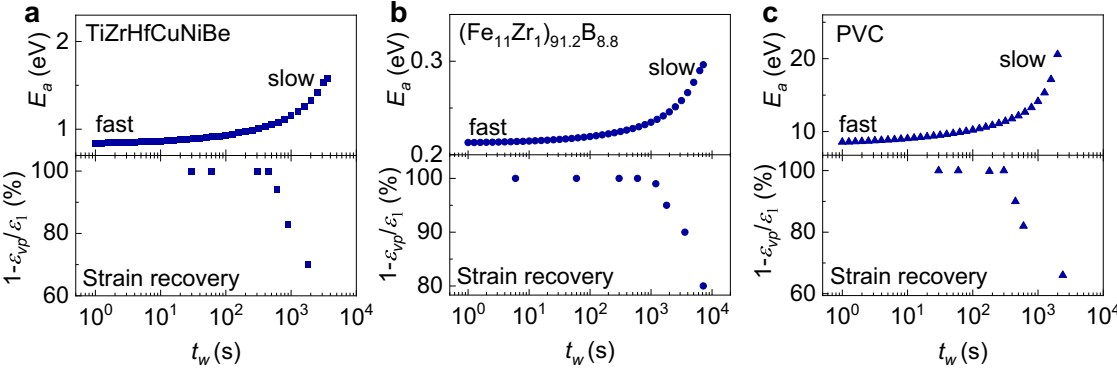

**Fig. 4 | Relaxation kinetics.** Evolution of (Top) the activation energy $E_a$ and (Bottom) the fraction of strain recovery ($1 - \varepsilon_{vp}/\varepsilon_1$) with $t_w$ for **a** TiZrHfCuNiBe MG, **b** $(Fe_{11}Zr_1)_{91.2}B_{8.8}$ MG, and **c** PVC polymer. The experimental condition for strain recovery tests: Ti-MG ($T_a = 593$ K, $\varepsilon_1 = 0.5\%$), Fe-MG ($T_a = 597$ K, $\varepsilon_1 = 0.4\%$), and PVC ($T_a = 318$ K, $\varepsilon_1 = 0.5\%$). (Calculation details for the activation energy and fraction of strain recovery are presented in sections 4 and 5 of Supplementary information). Source data are provided as a Source Data file.

which generate a more pronounced stress memory effect in the collective motion process (see Fig. 3). This aligns with the thriving of the Kovacs memory effect in a deep aging state characterized by a large activation entropy[15]. The herein reversible $\beta$ events also support the rational presence of the reversible faster relaxing element in the phenomenological Amir−Oreg−Imry (AOI) model[1,2], which has been widely used to explain the nonmonotonic relaxation phenomenon[2,22,23,54].

The classical Kovacs memory effect has a typical three-temperature protocol[5,6,19,25,34,55], say $T_0$, $T_2$ and $T_1$, where $T_0$ is the initial equilibrated temperature, $T_1$ and $T_2$ are annealing temperatures. The Kovacs memory effect happens only as $T_0 > T_2 > T_1$, while herein the stress memory effect happens only $\varepsilon_0 < \varepsilon_2 < \varepsilon_1$. Inspired by the theoretical works of Prados et al.[14,19,26], the contribution of the temperature perturbation to the glass system is alike to the stress/strain by considering $\beta E \propto \beta \sigma \varepsilon$. Here $\beta = \frac{1}{k_B T}$, and $k_B$ is the Boltzmann constant. $E$ represents the energy, which is basically proportional to $\sigma \varepsilon$[19]. Then, we can understand the effect of applied strain $\varepsilon$ on the glass system as a perturbation $\beta$ in the Kovacs memory effect. For instance, the low-to-high temperature annealing leads to a decrease in $\beta$, which is equal to the stimulation of high-to-low strain annealing in the protocol under a fixed temperature (Fig. 1a). The convergence of the two memory effects stems from their common structural origin, as the classical Kovacs memory effect also may arise from the recovery of reversible loosely packed regions[56,57].

In conclusion, we have observed a Kovacs-like stress memory effect when glasses are stimulated by high-to-low two-step strain stimulations. The stress memory only appears when the second-step strain is smaller than the first-step strain. Larger applied strain, longer stimulation time in the first stage, and higher temperature make the memory effect stronger. The runback of reversible motions of small atoms is the essential mobilizing mechanism of the memory effect, which is confirmed by the reconfiguration of short-range atomic packing structures.

## Methods
### Sample fabrication
The master alloys of $Ti_{16.7}Zr_{16.7}Hf_{16.7}Cu_{16.7}Ni_{16.7}Be_{16.7}$, $(Fe_{11}Zr_1)_{91.2}B_{8.8}$ (at%) were prepared by arc melting. The metallic glass ribbons with a thickness of approximately 30 μm were fabricated by the melt spinning with a tangent velocity of approximately 40 m/s. Polyvinyl chloride films with a thickness of approximately 150 μm were purchased from the manufacturer.

### Dynamic characterization
The dynamical mechanical behavior of the three glasses was measured using the dynamic thermomechanical analysis apparatus (TA-DMA Q800). The mechanical spectroscopy measurements were performed at a heating rate of 5 K/min with a driving frequency of 1 Hz.

### Stress-relaxation and strain-recovery measurements
Isothermal stress relaxation tests were performed on a TA DMA-Q800 instrument using one or two-step protocols (see Fig. 1a) at target temperatures. Subsequently, the strain recovery tests were conducted by unloading the stress after the relaxation period.

### Structural characterization
The local structural characterization was performed in China's Beijing synchrotron radiation facility. The in-situ experiments were conducted in the synchrotron X-ray beam at 3W1, which utilized monochromatic synchrotron radiation with a wavelength of 0.2065 Å (energy of 60.037 keV). The diffraction experiments were carried out in the Debye-Scherrer geometry. The layout of the experimental setup is shown in Supplementary Fig. 8a. The incident beam had a well-collimated cross-section of $0.8 \times 0.8$ mm². Diffraction patterns were collected using an iRay Mercu 1717HS image plate detector, which had a resolution of $3072 \times 3072$ pixels, with each pixel corresponding to a size of $139 \times 139$ μm². The detector was carefully positioned orthogonal to the x-ray beam, and the distance between the 2D detector and the sample was adjusted to about 280 mm in terms of covering a high-$Q$ range up to 16.5 Å$^{-1}$ (where $Q = 4\pi \sin\theta/\lambda$). $\theta$ is half the scattering angle, and $\lambda$ is the incident wavelength. The samples were exposed to the incident beam for 5 s during the experiments. Each total diffusion pattern was fully integrated to achieve the diffraction data and the Fourier transformations from $I(Q)$ to $G(r)$ were conducted using the PDFgetx3 program with the following equation[48]:

$$S(Q) = 1 + \frac{I_e(Q) - \left[\sum_{i=1}^{n} c_i f_i^2(Q)\right]}{\left[\sum_{i=1}^{n} c_i f_i^2(Q)\right]},$$

and $G(r) = \frac{2}{\pi} \int_0^{Q_{max}} Q(S(Q) - 1) \sin(rQ)dQ$.

Where $I_e(Q)$ represents the normalized elastically scattered intensity, and $r$ denotes the radial distance. $c_i$ and $f_i(Q)$ refer to the atomic concentration and the scattering factor of $i$th atomic in the tested alloy. The weighting factor of $i$-$j$ atoms pair was calculated

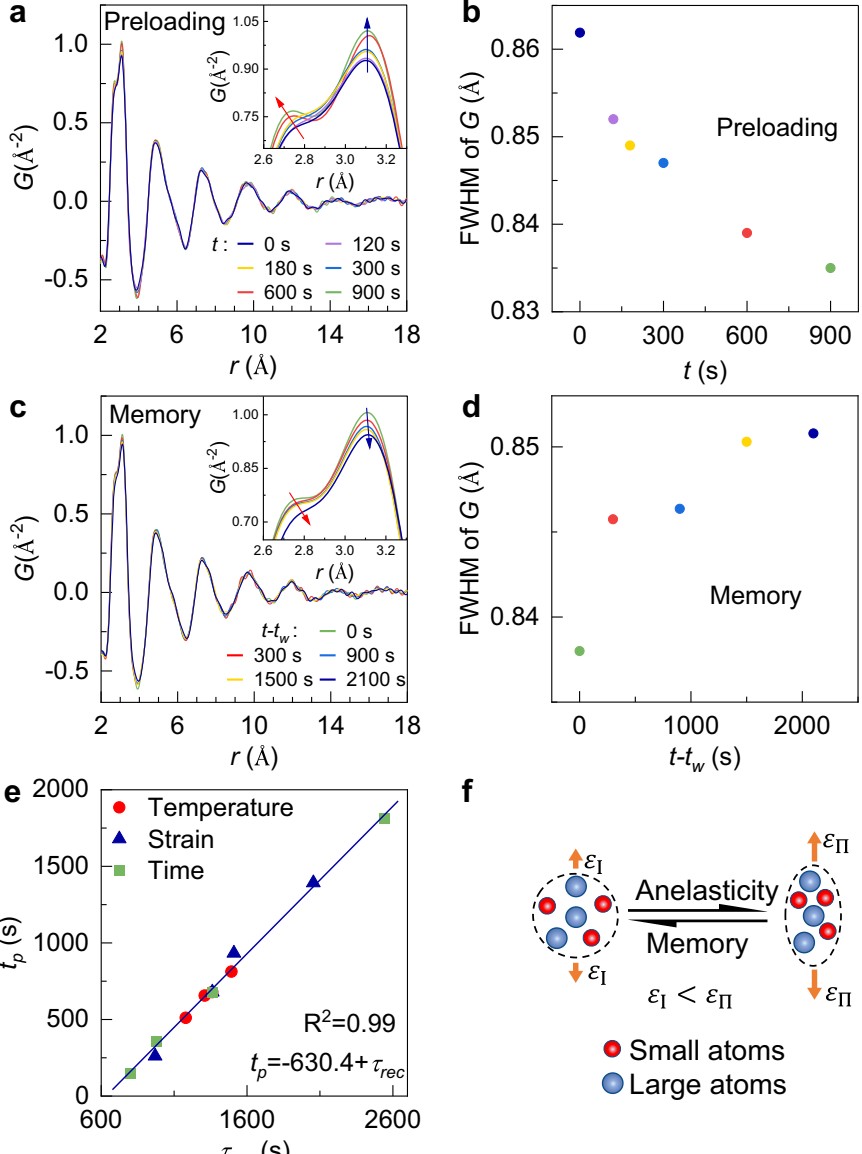

**Fig. 5 | Structural origin of stress memory effect for the TiZrHfCuNiBe high-entropy MG at 593 K. a** The reduced pair distribution function $G(r)$ in the stress decreasing stage during pre-loading at a fixed strain of 0.5% and **b** the corresponding evolution of the full width at half maximum (FWHM) of the first peak. **c** The evolution of $G(r)$ during the memory stage with $\varepsilon_2 = 0.3\%$ (after $\varepsilon_1 = 0.5\%$ and $t_w = 900$ s) and **d** the corresponding evolution of FWHM of the first peak. The insets show the zoomed-in data of the corresponding nearest neighbor atomic diffraction peaks. **e** The correlation between the memory peak time $t_p$ and the recovery

characteristic time $\tau_{rec}$. The square, triangle, and circular symbols represent data measured at different preloading variables of temperature, strain, and time, respectively. The data of peak time $t_p$ are taken from Fig. 2 and Supplementary Fig. 2. The blue solid straight line represents the linear fitting of $t_p = -630.4 + \tau_{rec}$, with a high regression coefficient of $R^2 = 0.99$. **f** The schematic illustration for the evolution of atomic packing structure for anelastic stress relaxation and recovery (memory) process. Source data are provided as a Source Data file.

according to the following manner[48]:

$$
w_{ij} = \begin{cases} \dfrac{2c_i c_j f_i(0) f_j(0)}{\left[\sum\limits_{k=1}^{n} c_k f_k(0)\right]^2}, i \neq j \\ \dfrac{c_i^2 f_i^2(0)}{\left[\sum\limits_{k=1}^{n} c_k f_k(0)\right]^2}, i = j \end{cases}.
$$

The specimens of amorphous ribbon were subjected to tension using the Linkam TST350 straining system. Supplementary Fig. 8b shows the loading protocol for tensile stress relaxation with two different applied strains ($\varepsilon_1 = 0.5\%, \varepsilon_1 = 0.3\%$) at a temperature of 593 K. The tensile strain was increased in increments of 200%/min, and the

gauge length of Linkam TST350 was 40 mm. The stress evolution corresponding to the two-step process is shown in Supplementary Fig. 8c.

### Reporting summary

Further information on research design is available in the Nature Portfolio Reporting Summary linked to this article.

## Data availability

All the necessary data that supports the findings of this article are provided in the main text and supplementary information. Source data are provided as a Source Data file. Additionally, the corresponding authors can provide the raw data upon request. Source data are provided with this paper.

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

## Acknowledgements

The X-ray pair distribution function measurements were performed at the 3W1 beamline, Beijing Synchrotron Radiation Facility (BSRF). This work was supported by the National Natural Science Foundation of China (NSFC 52201193, 52231006, 52201194, 92163108, 52001319), the National Key R&D Program of China (2018YFA0703600), the Natural Science Foundation of Ningbo City (No. 2022J310), 3315 Innovation Youth Talent in Ningbo City (2021A-123-G). E.P. acknowledge financial support from 'Proyecto PID2020-112975GB-I00 de investigación financiado por MCIN/AEI' and Generalitat de Catalunya AGAUR Grant No. 2021-SGR-00343.

## Author contributions

J.Q.W., L.J.S. and E.P. initiated and conceived the current work. Y.T. fabricate the samples, perform the stress relaxation experiments and analyze the data. L.L.F, F.C.L. Y.M.Y., G.M. and Y.T. perform the in-situ synchrotron X-ray experiments and analyze the data. All the authors discussed the results and contributed to writing the manuscript.

## Competing interests

The authors declare no competing interest.
