## [Peer Review File · Nature Communications]

Reviewer #1:
Remarks to the Author:

In the paper “Strain-driven Kovacs-like memory effect in glasses”, the authors present experimental results for a Kovacs-like memory effect for the stress response during a two-step strain perturbation. The authors argue that, despite the similarity with the original Kovacs effect, their results are unexpected because (after the waiting time window) the strain changes from “high” to “low” values, in contrast with the original Kovacs setup in which the temperature changes from “low” to “high” values.

I have several concerns about the paper, which could be summarised as follows: (i) implementation of the protocol, (ii) interpretation of the results, (iii) incomplete referencing of relevant work. I will address the three issues below, although (ii) and (iii) are closely related.

- (i) The Kovacs protocol, as originally designed by Kovacs, has three temperatures, say T_0 , T_1 and T_2 and thus two temperature jumps. First, the system is equilibrated at a “high” temperature T_0 and aged from this state during a waiting time t_w at a lower temperature T_1 . This isothermal relaxation at T_1 , which is slow in glassy systems, is interrupted at the waiting time, at which the temperature is abruptly changed to T_2 , in such a way that the instantaneous value of the relevant physical quantity—e.g. the enthalpy H —equals its equilibrium value at T_2 . The subsequent “Kovacs hump” for $t > t_w$, in which the physical quantity rebounds and passes through a maximum, shows that there are additional variables—aside from T and H —that characterise the dynamical state of the system.

It is the second temperature jump, from T_1 to T_2 , that the authors describe as a change from “low” to “high” value. It is important to remark, still, that T_2 is an intermediate temperature, $T_0 > T_2 > T_1$, but not any intermediate temperature: it must make the instantaneous value of the relevant physical quantity equal its equilibrium value at T_2 .

In this paper, the strain plays the role of the temperature whereas the stress plays the role of the enthalpy. The authors only give two strain values, ϵ_1 and ϵ_2 . The initial value of the strain is not explicitly given, but implicitly one understands that $\epsilon_0 = 0$. Assuming this, I have one concern about the experimental protocol. It seems that the authors consider the strain value after the waiting time as an independent variable; in fact, they consider both the cases $\epsilon_2 > \epsilon_1$ and $\epsilon_1 > \epsilon_2$. Of course, this can be done but it is not Kovacs’s protocol: in the Kovacs protocol, ϵ_2 only has one possible value, that determined (in the authors’ notation) that $\sigma(t = t_w) = \sigma_{eq}(\epsilon = \epsilon_2)$. So, unless I am not understanding it rightly, the authors are not actually implementing the Kovacs protocol in the system.

- (ii) The authors claim that the fact that the strain is changed from “high” to “low” values of the strain, i.e. $\epsilon_1 > \epsilon_2$ is essentially the opposite of the change from “low” to “high” values of the temperature in the Kovacs protocol, i.e. $T_2 > T_1$.

I believe that this claim is wrong. From my viewpoint, the two processes are completely alike.

First, from the authors setup, I would expect a Kovacs-like memory effect for $\epsilon_1 > \epsilon_2$, since the succession of strain values is $0 \rightarrow \epsilon_1 \rightarrow \epsilon_2$, in which ϵ_2 lies in between 0 and ϵ_1 , in complete analogy with T_2 lying in between

T_0 and T_1 . In the Kovacs setup, if we considered $T_2 < T_1$, the Kovacs effect would not emerge, since the enthalpy at t_w is above its equilibrium value at T_1 , which would be larger than its equilibrium value at T_2 .

Second, the canonical distribution is proportional to $\exp(-\beta E)$, where $\beta = (k_B T)^{-1}$ and E is the energy, which is basically proportional to $\epsilon \sigma$. Therefore, $\beta E \propto \beta \epsilon \sigma$; so the perturbations in ϵ should be compared with perturbations in β , not in the temperature. And, indeed, $\beta_0 \rightarrow \beta_1 \rightarrow \beta_2$ has the same structure as $\epsilon_0 \rightarrow \epsilon_1 \rightarrow \epsilon_2$; in the Kovacs protocol one has $\beta_0 < \beta_2 < \beta_1$, in complete analogy with the protocol employed here, $\epsilon_0 < \epsilon_2 < \epsilon_1$.

I would also like to point out that, in general, one would also expect a Kovacs-like hump or rebound if $\beta_0 > \beta_2 > \beta_1$, i.e. if $T_0 < T_2 < T_1$, the temperature is first increased from T_0 to T_1 and afterwards, at $t = t_w$, suddenly decreased to the intermediate value T_2 . The key points are always that (i) T_2 lies in between T_0 and T_1 , and (ii) T_2 is chosen in such a way that the instantaneous value of the quantity of interest at t_w equals the equilibrium value at T_2 .

(iii) There are some papers that I think are relevant for the authors's analysis of the Kovacs effect. In particular, the theoretical studies contain some key results that may help the authors to improve their understanding of the system response.

- PRL 92, 045504 (2004): a seminal simulation study of the Kovacs effect, e.g. in Fig.1 the authors can find the typical three-temperature protocol ($T_h = T_0$, T_1 is the same, $T = T_2$).
- Soft Matter 6, 3065–3073 (2010): a theoretical study of the Kovacs effect through the lens of non-equilibrium thermodynamics, with quite a complete analysis of the simulation results in PRL 92, 045504 (2004).
- J. Stat. Mech. (2010) P02009: another theoretical study of the Kovacs effect, but through the lens of stochastic processes. General results for the shape of the Kovacs hump are derived in linear response, basically assuming that the equilibrium distribution has the canonical form.
- PRL 112, 198001 (2014): a theoretical study of a Kovacs-like effect in granular fluids that shows that a complex energy landscape is not necessary for the emergence of the Kovacs effect. (The authors already cite some papers on memory effects on granular systems, Refs. [21] and [23].)
- PRL 119, 188001 (2017): building on the previous paper, a theoretical study of a Kovacs-like effect in active matter; it also proposes a generalisation of the results in J. Stat. Mech. (2010) P02009 to non-linear response.
- PRL 125, 058001 (2020) (already cited): this paper analyses a Kovacs-like effect in a protein construct, and somehow it may be considered as the "negative" image of the work presented here, presenting results of length response under a protocol changing the applied tension (vs.

the "positive" image here, stress response under a protocol changing the strain). I believe this might be highlighted.

It may well be possible that some of the behaviours found by the authors and shown in Figs. 2–5 could be understood in the light of the theoretical studies, and also that the interpretation of the results could be improved by comparing them with the numerical and experimental work referred above.

As a whole, once a thorough revision of the results presented in the paper is done, I believe that the paper contains enough new results to be published in a good journal. But the results are quite standard, neither giving essential new information about the emergence of the Kovacs effect in glassy systems nor opening new perspectives in the field of memory effects, so a journal different from Nature Communications should be considered.

Reviewer #2:

Remarks to the Author:

The paper reports a very interesting phenomenon, a peculiar stress memory effect upon two steps strain stimulation. This occurs when the strain stimulation jumps from a higher to lower state states. The authors discuss their findings in terms of high collective atomic motions, and they present an atomic model to explain the phenomenon. Three different amorphous systems, two metallic glasses and an amorphous polymer have shown this memory effect, supporting according to the authors the universality of this phenomenon in different types of glasses.

The paper is indeed very interesting and well written and merits publication.

However, there are some points the authors need to address before the paper can be accepted for publication.

Comments

1. The authors should provide more information about their experiments especially in the case of the ins-situ synchrotron studies and subsequent data processing and analysis. Have the authors considered the orientation of the strain on the on the volume of material analysed (see for example "Strain distribution in Zr_{64.13}Cu_{15.75}Ni_{10.12}Al₁₀ bulk metallic glass investigated by in situ tensile tests under synchrotron radiation", M. Stoica, et al, J Appl Phys 2008).

2. in page 8 the paper reads: "Close inspections indicate that the first diffraction peaks split..." The term diffraction peaks is not appropriate here; since the authors refer the real space pair distribution functions and the reciprocal space diffraction patterns.

3. The PDF data shown in fig 5 indicate some structural rearrangement in the first nearest neighbour shell (first PDF peak). I wonder if the corresponding diffraction data (and any changes in the first diffraction peak) can provide further information on the structural evolution during at the different stages of strain.

Reviewer #3:

Remarks to the Author:

In the current manuscript, Tong et al. present the results of their experimental study on a mechanical memory effect, referred to as a "Kovacs-like memory effect". This study involves the application of strain loading on assorted glass materials, and after a designated stress relaxation period, the strain is diminished to a lower level. Notably, an increase in the stress level is observed before it ultimately decreases. The authors scrutinize this phenomenon through the use of in-situ synchrotron X-ray experiments and relaxation kinetics. This comprehensive examination of the seemingly universal phenomenon is commendable as it spanned across different types of glasses, namely two metallic glasses and a polymer glass (PVC). This paper offers substantial insights and is well-structured. I'd recommend its publication post proper revisions on the following points:

English Language Refinement:

Page 2: The sentence "One kernel is the complex relaxation behaviors derived from nonequilibrium thermodynamics" requires revision. The term 'kernel' seems incongruous in this context and there are grammatical inconsistencies.

Page 2: The phrase "For structural glasses, recent results suggest that the stress field is equal to the thermal effect for triggering the flow" lacks clarity. The implication of "equal" in this context is ambiguous.

Page 2: The assertion that the Kovacs-like memory effect is valid for structural glasses could benefit from additional specificity concerning the type of materials where this memory effect might be observed.

Reference Provision:

Page 4: The authors should cite a reference for the Kohlrausch-Williams-Watts (KWW) expression.

Clarification of Memory Effect Understanding:

The understanding of the memory effect appears to hinge on the anelastic deformation process, colloquially known as beta relaxation. The X-ray experiments results, particularly the $G(r)$ and the evolution of the Full Width at Half Maximum (FWHM) of its first peak, suggest a correlation with the memory effect. However, the types of atomic displacements associated with the stress level rise are not clearly outlined.

Distinction of Anelastic and Visco-plastic Deformation:

There is an apparent lack of clarity on how the authors differentiated between anelastic and visco-plastic deformation in the three different types of glasses during the relaxation process in this study.

Link to Theoretical Discussion:

In the introduction, the authors refer to theories addressing the relaxation process in glasses and the memory effect (for instance, ref. 15). Nevertheless, in the discussion section, it remains unclear whether the obtained data supports any specific theoretical perspectives of the memory effect or if these results are extraneous to the discussions of the Kovacs memory effect. The authors assert near the conclusion that both memory effects originate from the same structural source, but it is unclear if they can be described within the same theoretical framework.

Dear Reviewers,

Thank you very much for your suggestions and comments. We have carefully revised the manuscript. Some additional experiments are done to clarify your concerns. All the concerns from the reviewers should have been addressed properly. The modifications in main texts have been marked using red color. The one-to-one responses can be found below. We believe that the revised manuscript should be suitable for publication. Thank you very much!

Point-by-point Responses:

Part I: for Reviewer 1

In the paper “Strain-driven Kovacs-like memory effect in glasses”, the authors present experimental results for a Kovacs-like memory effect for the stress response during a two-step strain perturbation. The authors argue that, despite the similarity with the original Kovacs effect, their results are unexpected because (after the waiting time window) the strain changes from “high” to “low” values, in contrast with the original Kovacs setup in which the temperature changes from “low” to “high” values.

I have several concerns about the paper, which could be summarised as follows: (i) implementation of the protocol, (ii) interpretation of the results, (iii) incomplete referencing of relevant work. I will address the three issues below, although (ii) and (iii) are closely related.

Response: We would like to express our sincere gratitude for your valuable suggestions and comments on our work. The insightful points help improving the discussions of our manuscript. We have carefully revised the manuscript based on your suggestions. The one-to-one responses can be found below.

Question 1. The Kovacs protocol, as originally designed by Kovacs, has three temperatures, say T_0 , T_1 and T_2 and thus two temperature jumps. First, the system is equilibrated at a “high” temperature T_0 and aged from this state during a waiting time t_w at a lower temperature T_1 . This isothermal relaxation at T_1 , which is slow in glassy systems, is interrupted at the waiting time, at which the temperature is abruptly changed to T_2 , in such a way that the instantaneous value of the relevant physical quantity—e.g. the enthalpy H —equals its equilibrium value at T_2 . The subsequent “Kovacs hump” for $t > t_w$, in which the physical quantity rebounds and passes through a maximum, shows that there are additional variables—aside from T and H —that characterise the dynamical state of the system.

It is the second temperature jump, from T_1 to T_2 , that the authors describe as a change from “low” to “high” value. It is important to remark, still, that T_2 is an intermediate temperature, $T_0 > T_2 > T_1$, but not any intermediate temperature: it must make the instantaneous value of the relevant physical quantity equal its equilibrium value at T_2 .

In this paper, the strain plays the role of the temperature whereas the stress plays the role of the enthalpy. The authors only give two strain values, ϵ_1 and ϵ_2 , The initial value of the strain is not explicitly given, but implicitly one understands that $\epsilon_0 = 0$. Assuming this, I have one concern about

the experimental protocol. It seems that the authors consider the strain value after the waiting time as an independent variable; in fact, they consider both the cases $\epsilon_2 > \epsilon_1$ and $\epsilon_1 > \epsilon_2$. Of course, this can be done but it is not Kovacs's protocol: in the Kovacs protocol, ϵ_2 only has one possible value, that determined (in the authors' notation) that $\sigma(t = t_w) = \sigma_{eq}(\epsilon = \epsilon_2)$. So, unless I am not understanding it rightly, the authors are not actually implementing the Kovacs protocol in the system.

Response: As the Reviewer has noted, our protocol involves three strain variables, ϵ_0 , ϵ_1 and ϵ_2 , as depicted in the protocol Fig. R1. Here we set $\epsilon_0 = 0$, which is analogous to the T_0 in the classical Kovacs protocol. In the classical Kovacs protocol, T_0 is not discussed very much because it is usually referred to the glass transition temperature T_g where the glass is frozen in. This is like what we do in the experiments: the initial state of glass always starts from the zero-strain $\epsilon_0 = 0$. Thus, here we focus on discussing about the influence of the other two strains and annealing time on the memory strength.

Fig. R1. Three-step strain annealing protocol for stress memory effect. **a** The schematic of tensile stress relaxation with an initial strain $\epsilon_0 = 0$ and two different applied strains (ϵ_1 , ϵ_2). The preloading time is t_w . F represents axial tensile force. See also revised Fig.1(a) of the main text.

Fig. R2. The two-jumping experiments designed by the Kovacs in 1958, copy from the literature¹

About the thermal protocols [that $\sigma(t = t_w) = \sigma_{eq}(\epsilon = \epsilon_2)$], the Reviewer referred to a specific case that was done by Kovacs. There are many other common protocols to trigger the memory effect, which were also introduced by Kovacs, as can be found below.

Kovacs initially designed the crossover experiments in 1958 (*J. Polym. Sci.* 30, 131-147 (1958)), as depicted in Fig. R2. This involved setting up a series of annealing routes with two different temperatures, including *high-to-low* temperature annealing (a, b) and *low-to-high* temperature annealing (d, e), as shown in the inset figure. Only for routes d and e, the volume of the glass exhibits a non-monotonic evolution or crossover with time. Similar cases were reported by Kovacs in 1963 (*Adv. Polym. Sci.* 3, 394-507 (1963)).

Later, Kovacs named this crossover as the memory effect in 1979 (*J. Polym. Sci.* 17, 1097-1162 (1979)), which involves a three-step temperature history: a down jump is made from equilibrium temperature T_0 to T_1 , where the sample is aged for different sufficient times (t_1). As the Reviewer noted, the enthalpy state at T_1 should be equal to the equilibrium state at T_2 , as shown in *routes B, C, D, and E* in the left figure of Fig. R3. This is why the reviewer thought “ ϵ_2 only has one possible value, that determined (in the authors’ notation) that $\sigma(t = t_w) = \sigma_{eq}(\epsilon = \epsilon_2)$ ”.

However, the memory effect is NOT limited to that specific protocol. As shown in the right figure of Fig. R3, Kovacs also introduced the memory effect starting from the non-equilibrium state at T_2 , as depicted by Routes *B', D'* and *E'*. Kovacs named all these crossover evolutions as the memory effect, as marked in the figure caption. Such crossover from the non-equilibrium state has been widely studied and referred to as the Kovacs memory effect, for instance, *PRL* 92, 045504 (2004), see Fig. R4. Thus, it is reasonable to name the non-monotonous stress relaxation as the Kovacs-like memory effect, despite the $\sigma(t = t_w) \neq \sigma_{eq}(\epsilon = \epsilon_2)$.

Fig. R3. The *memory effect* experiments performed by the Kovacs. The protocol has three temperatures T_0 , T_1 , and T_2 . **Left:** Routes *B, C, D,* and *E* denote the crossover from the equilibrium state. **Right:** the crossovers from the equilibrium state (Route *C'*) and the non-equilibrium states (Routes *B', D'* and *E'*)

Fig. R4. Crossover from the non-equilibrium state in an aging molecular liquid (*Phys. Rev. Lett.* 92, 045504 (2004)).

Question 2. The authors claim that the fact that the strain is changed from “high” to “low” values of the strain, i.e. $\epsilon_1 > \epsilon_2$ is essentially the opposite of the change from “low” to “high” values of the temperature in the Kovacs protocol, i.e. $T_2 > T_1$. I believe that this claim is wrong. From my viewpoint, the two processes are completely alike.

First, from the authors setup, I would expect a Kovacs-like memory effect for $\epsilon_1 > \epsilon_2$, since the succession of strain values is $0 \rightarrow \epsilon_1 \rightarrow \epsilon_2$, in which ϵ_2 lies in between 0 and ϵ_1 , in complete analogy with T_2 lying in between T_0 and T_1 . In the Kovacs setup, if we considered $T_2 < T_1$, the Kovacs effect would not emerge, since the enthalpy at t_w is above its equilibrium value at T_1 , which would be larger than its equilibrium value at T_2 .

Second, the canonical distribution is proportional to $\exp(-\beta E)$, where $\beta = (k_B T)^{-1}$ and E is the energy, which is basically proportional to $\epsilon \sigma$. Therefore, $\beta E \propto \beta \epsilon \sigma$; so the perturbations in ϵ should be compared with perturbations in β , not in the temperature.

And, indeed, $\beta_0 \rightarrow \beta_1 \rightarrow \beta_2$ has the same structure as $\epsilon_0 \rightarrow \epsilon_1 \rightarrow \epsilon_2$; in the Kovacs protocol one has $\beta_0 < \beta_2 < \beta_1$, in complete analogy with the protocol employed here, $\epsilon_0 < \epsilon_2 < \epsilon_1$.

I would also like to point out that, in general, one would also expect a Kovacs-like hump or rebound if $\beta_0 > \beta_2 > \beta_1$, i.e. if $T_0 < T_2 < T_1$, the temperature is first increased from T_0 to T_1 and afterwards, at $t = t_w$, suddenly decreased to the intermediate value T_2 . The key points are always that (i) T_2 lies in between T_0 and T_1 , and (ii) T_2 is chosen in such a way that the instantaneous value of the quantity of interest at t_w equals the equilibrium value at T_2 .

Response: Thank you very much for your valuable suggestion! It is smart to define the parameter $\beta = (k_B T)^{-1}$ in order to compare the roles of temperature, stress and strain in a thermal activation process in form of $\exp(-\beta \epsilon \sigma)$. We have updated the discussion part accordingly.

“The classical Kovacs memory effect has a typical three-temperature protocol^{5, 6, 19, 25, 34, 55}, say T_0 , T_2 and T_1 , where T_0 is the initial equilibrated temperature, T_1 and T_2 are annealing temperatures. The Kovacs memory effect happens only as $T_0 > T_2 > T_1$, while herein the stress memory effect happens only $\varepsilon_0 < \varepsilon_2 < \varepsilon_1$. Inspired by the theoretical works of Prados *et al.*^{14, 19, 26}, the contribution of the temperature perturbation to the glass system is alike to the stress/strain by considering $\beta E \propto \beta \sigma \cdot \varepsilon$. Here $\beta = \frac{1}{k_B T}$, and k_B is the Boltzmann constant. E represents the energy, which is basically proportional to $\sigma \cdot \varepsilon$ ¹⁹. Then, we can understand the effect of applied strain ε on the glass system as a perturbation β in the Kovacs memory effect. For instance, the *low-to-high* temperature annealing leads to a decrease in β , which is equal to the stimulation of *high-to-low* strain annealing in the protocol under a fixed temperature (Fig. 1a). The convergence of the two memory effects stems from their common structural origin, as the classical Kovacs memory effect also may arise from the recovery of reversible loosely packed regions^{56, 57}.”

Based on your inference regarding the emergence of a Kovacs-like hump with two temperature jumps ($T_0 < T_2 < T_1$), we conducted stress relaxation tests with two temperature jumps, as illustrated in Fig. R5a. Under a fixed strain of $\varepsilon_f=0.4\%$, we first subjected the sample to strain aging at $T_1=613$ K for $t_w=600$ s (starting from $T_0=300$ K). The stress exhibited a monotonic decay evolution during this stage, as shown in Fig. R5b. Subsequently, we decreased the temperature to $T_2=593$ K, and the stress displayed a non-monotonic evolution, increasing first and then decreasing with time (see Fig. R5c).

Fig. R5. Another three-temperature protocol in strain annealing also triggers the stress memory effect. **a** The schematic of tensile stress relaxation with three different applied temperatures (T_0 , T_1 , T_2) with $T_0 = 300$ K, $T_1 = 613$ K, and $T_2 = 593$ K; **b** The nominal stress evolution at $T_1 = 613$ K; **c** The stress memory phenomenon at 593 K.

Question 3. There are some papers that I think are relevant for the authors’s analysis of the Kovacs effect. In particular, the theoretical studies contain some key results that may help the authors to improve their understanding of the system response.

– PRL 92, 045504 (2004): a seminal simulation study of the Kovacs effect, e.g. in Fig.1 the authors can find the typical three-temperature protocol ($T_h = T_0$, T_1 is the same, $T = T_2$).

– Soft Matter 6, 3065–3073 (2010): a theoretical study of the Kovacs effect through the lens of non-equilibrium thermodynamics, with quite a complete analysis of the simulation results in PRL 92, 045504 (2004).

– J. Stat. Mech. (2010) P02009: another theoretical study of the Kovacs effect, but through the lens of stochastic processes. General results for the shape of the Kovacs hump are derived in linear response, basically assuming that the equilibrium distribution has the canonical form.

– PRL 112, 198001 (2014): a theoretical study of a Kovacs-like effect in granular fluids that shows that a complex energy landscape is not necessary for the emergence of the Kovacs effect. (The authors already cite some papers on memory effects on granular systems, Refs. [21] and [23].)

– PRL 119, 188001 (2017): building on the previous paper, a theoretical study of a Kovacs-like effect in the active matter; it also proposes a generalization of the results in J. Stat. Mech. (2010) P02009 to non-linear response.

– PRL 125, 058001 (2020) (already cited): this paper analyses a Kovacs-like effect in a protein construct, and somehow it may be considered as the “negative” image of the work presented here, presenting results of length response under a protocol changing the applied tension (vs. the “positive” image here, stress response under a protocol changing the strain). I believe this might be highlighted.

It may well be possible that some of the behaviors found by the authors and shown in Figs. 2–5 could be understood in the light of the theoretical studies, and also that the interpretation of the results could be improved by comparing them with the numerical and experimental work referred to above.

Response: Thanks for your professional suggestions.

First, the above-mentioned articles are highly beneficial in enhancing the theoretical aspects of our work, and we have duly cited all of them in the modified version of the manuscript.

Second, compare our stress memory effect to a Kovacs-like effect in a protein construct (PRL 125, 058001 (2020)). Regarding the *high-to-low* ($f_1=50$ pN to $f_2=9$ pN) iso-stress stimulation (as shown in Fig. R6b), a monotonous logarithmic extension has been observed. However, it is important to note that the initial stress, $f_0=0$ pN, was neglected in their analysis (see Fig. R6b). To further validate our speculation, we conducted similar two-step stress creep experiments in TiZrHfCuNiBe metallic glasses, as shown in Fig. R6d. Subjecting the MG to the three stresses σ_0 , σ_1 , and σ_2 with a relation $\sigma_0(=0)$ and $\sigma_1 > \sigma_2$, we can see the strain shows non-monotonous evolution, decrease first and then increase, if the loading time is long enough. Comparing figures R6c and e suggests that the data in fig. R6c is alike to the beginning part in fig. R6e, as marked by the dashed box.

For further verification, we performed three-step creep with four stresses σ_0 , σ_1 , σ_2 and σ_3 by setting the $\sigma_0 = 0$, $\sigma_1 > \sigma_3 > \sigma_2$, as shown in the Fig. R7d. We can see the last-step strain shows the *increase-decrease-increase* evolution. Comparing Figs.R7b and 7e suggests that their Kovacs-like effect (b) is alike to the partial zoom of our memory effect (e), as marked by the dashed

box. In summary, the Kovacs-like effect in a protein construct are similar with the metallic glass/polymer glass, but not “negative” result.

Fig. R6. a Experimental setup: A polymer consisting of multiple NFL IDRs joined by disulfide bonds is stretched with a force f while its extension L is tracked. Stuck beads are tracked to remove drift. **b** Example force-quench experiment on a single polymer: at $t=0$, the force was decreased from $f_1=50$ to $f_2=9$ pN, resulting in a rapid elastic response followed by a slow logarithmic relaxation (inset). **c** Typical force dependence of the logarithmic relaxation after quenching from $f_1=50$ pN, plotted as the compaction, $\Delta L \equiv L(t) - L(t_0)$, after a reference time $t_0=1$ s. At higher f_2 (labeled, in pN), the relaxation slows due to hindering of chain shortening by tension. Data points and error bars are the mean and standard error of the mean after logarithmic binning in time (error bars are smaller than points) Copy from Fig.1 of PRL 125, 058001 (2020); **d-e The experimental validations in TiZrHfCuNiBe metallic glass:** **d** The schematic of tensile creep protocol with two different stress (σ_1 , σ_2), and the preloading time is t_w ; **e** Data from the second-step creep, represented on a logarithmic time scale, the raw data of TiZrHfCuNiBe metallic glass under $\sigma_1=300$ MPa, $t_w=900$ s, and $\sigma_2=50, 100, 200, 250, 400$ MPa at a fixed annealing temperature 593 K.

Fig. R7. **a** Typical two-step experiment: The force was initially $f_1=60$ pN, then held at $f_2=7$ pN for $t_w=10$ s, then increased to $f_3=19$ pN. **b** Detail of extension dynamics at f_3 from the data in **a**, showing a nonmonotonic change. Data points and error bars are the mean and standard error of the mean after logarithmic binning in time (error bars are smaller than points). **c** Cartoon of heterogeneous dynamics within a single IDR domain that results in the Kovacs hump: Incubation at f_2 for t_w (left) allows folding offset segments, but is not long enough to allow folding of slow segments. Application of the higher force f_3 causes the unfolding of some fast segments, leading to the increase, $L_2>L_1$. At long times, the slow segments finally fold, causing the slow decrease, $L_3<L_2$. Copy from Fig.2 of PRL 125, 058001 (2020); **The experimental validations in TiZrHfCuNiBe metallic glass:** **d** Three-step protocol for the tensile creep. **e** The effect of σ_1 on the creep strain of TiZrHfCuNiBe metallic glass with the fixed $\sigma_0 = 0$ MPa, $\sigma_2 = 0.01$ MPa, $\sigma_3 = 100$ MPa, $t_{w1}=900$ s, and $t_{w2}=90$ s at 593 K.

Question 4. As a whole, once a thorough revision of the results presented in the paper is done, I believe that the paper contains enough new results to be published in a good journal. But the results are quite standard, neither giving essential new information about the emergence of the Kovacs effect in glassy systems nor opening new perspectives in the field of memory effects, so a journal different from Nature Communications should be considered.

Response: We would like to express our acknowledgements for your insights into the intimate relation between the temperature and stress/strain in inducing the memory effect, which makes the

theoretical discussions on understanding the stress memory effect more comprehensively. Based on the above responses, the reviewer should have noticed that this is the first time to report such an abnormal relaxation phenomena in metallic glasses and polymer glasses, even though there are similar stress stimulation protocols in colloidal, proteins and simulation works on disordered systems. The thermodynamic and structural analysis also contribute insightful understandings on the non-equilibrium behaviors of glasses. Thus, we believe this novel, sound and insightful work should be of wide interests and be valuable for publication in Nature Communications.

On the other hand, the roles of temperature and stress in stimulating the flow behavior of glasses have attracted broad interests in the community. It was reported that increasing temperature and increasing the strain are equivalent in driving glass to flow [*PRL 104, 205701 (2010)*]. Thus, the roles of temperature and stress in triggering abnormal relaxation and flow are different. This is interesting and will be studied in details in future.

For practical applications, the glasses are usually utilized as load-bearing structural materials. Overcoming the stress drop is very important to keep the stability of the structure. According to our observations, the implementation of two-step strain stimulations is a novel regulatory method for avoiding stress drop and promoting the application of glass materials.

Part II: for Reviewer 2

The paper reports a very interesting phenomenon, a peculiar stress memory effect upon two steps strain stimulation. This occurs when the strain stimulation jumps from a higher to lower state states. The authors discuss their findings in terms of high collective atomic motions, and they present an atomic model to explain the phenomenon. Three different amorphous systems, two metallic glasses, and an amorphous polymer have shown this memory effect, supporting according to the authors the universality of this phenomenon in different types of glasses.

The paper is indeed very interesting and well written and merits publication.

However, there are some points the authors need to address before the paper can be accepted for publication.

Response: We are grateful for your positive evaluation of our work.

Comments

1. The authors should provide more information about their experiments especially in the case of the in-situ synchrotron studies and subsequent data processing and analysis. Have the authors considered the orientation of the strain on the on the volume of material analysed (see for example “Strain distribution in $Zr_{64.13}Cu_{15.75}Ni_{10.12}Al_{10}$ bulk metallic glass investigated by in situ tensile tests under synchrotron radiation”, M. Stoica, et al, J Appl Phys 2008).

Response: Thank you for your professional feedback. We have incorporated detailed information about the *in-situ* synchrotron studies and the subsequent data processing in the section of the **Methods**, which is highlighted in red color. This information can also be found in the subsequent section:

“The local structural characterization was performed in China's Beijing synchrotron radiation facility. The *in-situ* experiments were conducted in the synchrotron X-ray beam at 3W1, which utilized monochromatic synchrotron radiation with a wavelength of 0.2065 Å (energy of 60.037 keV). The diffraction experiments were carried out in the Debye-Scherrer geometry. The layout of the experimental setup is shown in Supplementary Fig. 8a. The incident beam had a well-collimated cross-section of 0.8×0.8 mm². Diffusion patterns were collected using an iRay Mercu 1717HS image plate detector, which had a resolution of 3072×3072 pixels, with each pixel corresponding to a size of 139×139 μm². The detector was carefully positioned orthogonal to the x-ray beam, and the distance between the 2D detector and the sample was about 280 mm in terms of covering a high- Q range up to 16.5 Å⁻¹ (where $Q = 4\pi\sin\theta/\lambda$). r is the distance in real space, θ is half the scattering angle, and λ is the incident wavelength. The samples were exposed to the incident beam for 5 seconds during the experiments. Each total diffraction pattern was fully integrated to achieve the diffraction data and the Fourier transformations from $I(Q)$ to $G(r)$ were conducted using the PDFgetx3 program with the following equation⁴⁸: $S(Q) = 1 + \frac{I_e(Q) - [\sum_{i=1}^n c_i f_i^2(Q)]}{[\sum_{i=1}^n c_i f_i^2(Q)]}$, and $G(r) =$

$$\frac{2}{\pi} \int_0^{Q_{max}} Q(S(Q) - 1) \sin(rQ) dQ.$$

Where $I_e(Q)$ represents the normalized elastically scattered intensity, and r denotes the radial distance. c_i and $f_i(Q)$ refer to the atomic concentration and the scattering factor of i th atomic in the tested alloy. The weighting factor of i - j atoms pair was calculated according to the following

$$\text{manner}^{48}: w_{ij} = \begin{cases} \frac{2c_i c_j f_i(0) f_j(0)}{[\sum_{k=1}^n c_k f_k(0)]^2}, & i \neq j \\ \frac{c_i^2 f_i^2(0)}{[\sum_{k=1}^n c_k f_k(0)]^2}, & i = j \end{cases}.$$

The specimens of amorphous ribbon were subjected to tension using the Linkam TST350 straining system. Supplementary Fig. 8b shows the loading protocol for tensile stress relaxation with two different applied strains ($\varepsilon_1 = 0.5\%$, $\varepsilon_2 = 0.3\%$) at a temperature of 593 K. The tensile strain was increased in increments of 200%/min, and the gauge length of Linkam TST350 was 40 mm. The stress evolution corresponding to the two-step process is shown in Supplementary Fig. 8c.”

We have carefully read the paper by M. Stoica et al. (*J. Appl. Phys.* 104, 013522 (2008)) which provides a detailed and professional analysis of *in-situ* tensile tests conducted under synchrotron radiation, and we added a citation to this paper. The paper explains how the stress influences the volume of the sample, leading to anisotropy. This anisotropy is manifested by distinct positions of the first diffraction peaks at various azimuth angles φ . The anisotropy or asymmetry of the diffraction pattern shows an increase with increasing load. For our data, we have not taken into account the orientation of the applied strain on the analyzed material volume, because the stress in our work doesn't induce strong asymmetry. In our *in-situ* two-step stress relaxation tests, the maximum loading strain is 0.5% with a maximum stress of around 225 MPa. This stress doesn't induce significant asymmetry in the material, as an example the stress in M. Stoica *et al* work is as higher as 800 MPa and 1400 MPa. Fig. R8a shows an example of the raw data of the 2D diffraction pattern obtained for TiZrHfCuNiBe metallic glass under the iso-strain loading with a strain of 0.5% at 597 K. The corresponding two-dimensional contours of the diffraction intensity are shown in Fig. R8b, where the intensity of diffraction peaks is highlighted and exhibits a vertical linear relationship with the 2θ axis or a parallel linear relationship with the φ axis. The symmetric circular diffraction pattern suggests the negligible effect of the orientation of the strain on the volume of material.

Fig. R8. a 2D diffraction pattern of TiZrHfCuNiBe metallic glass under the iso-strain loading with a strain of 0.5% at 597 K. The white solid arrows denote the tensile deformation. **b** The two-dimensional contours of the diffraction intensity with the variables of 2θ and φ . The dashed line serves as a visual guide to aid in interpretation.

2. in page 8 the paper reads: “Close inspections indicate that the first diffraction peaks split....” The term diffraction peaks is not appropriate here; since the authors refer the real space pair distribution functions and the reciprocal space diffraction patterns.

Response: Thank you for your professional advice. We change the term “diffraction peaks” to “the first nearest neighbor shells” in the modified version.

3. The PDF data shown in fig 5 indicate some structural rearrangement in the first nearest neighbor shell (first PDF peak). I wonder if the corresponding diffraction data (and any changes in the first diffraction peak) can provide further information on the structural evolution during at the different stages of strain.

Response: We appreciate your professional suggestion. We plot curves of the diffraction data $S(Q)$ during the different stages of strain, as shown in Fig. R9. Figs. R9a and c show the structure factor $S(Q)$ of the TiZrHfCuNiBe metallic glass for the preloading and memory stages, respectively. The inset shows the enlargement of the change of the corresponding first diffraction peak Q_1 . During the preloading stress relaxation stage with an iso-strain of 0.5%, we can see the position of the Q_1 shifts toward the right slightly with the relaxation time, while the peak height increases (Figs. R9b). This sharpening phenomenon has been interpreted as an indication of enhanced local ordering during the preloading relaxation, which is consistent with findings from previous studies (*Acta Mater.* 241, 118376 (2022). *Mater. Res. Lett.* 11, 547-555 (2023). *Nat. Commun.* 7, 10344 (2016)). For example, Ge *et al.* observed the sharpening of the first diffraction peak (indicative of ordering) in $Zr_{50}Cu_{40}Al_{10}$ metallic glass under winding annealing conditions (*Mater. Res. Lett.* 11, 547-555 (2023)). Similarly, Ruta *et al.* reported continuous sharpening of the first diffraction peak (indicative of ordering)

during the single-step relaxation of Pd₇₇Si_{16.5}Cu_{6.5} metallic glass (*Nat. Commun.* 7, 10344 (2016)). As the applied stress switch to lower strain ε_2 , triggering the memory effect (stress increase), the position of the Q_1 shifts toward the left slightly with time, and the peak height decreases (Figs. R9d). This suggests the sample returned to a disordered state, which is in accordance with the rejuvenating structural evaluation in metallic glass (*Mater. Res. Lett.* 11, 547-555 (2023)).

Fig. R9. The structure evolution for stress memory effect in TiZrHfCuNiBe metallic glass. **a** The structure factor patterns $S(Q)$ patterns and **b** the corresponding peak height of the first diffraction peak Q_1 during the preloading stage (stress decay); **c** The structure factor patterns $S(Q)$ patterns and **d** the corresponding peak height of the first diffraction peak Q_1 during the memory stage (stress increase). The stress evolution curves are shown in Supplementary Fig. 8c.

Part III: for Reviewer 3

In the current manuscript, Tong et al. present the results of their experimental study on a mechanical memory effect, referred to as a "Kovacs-like memory effect". This study involves the application of strain loading on assorted glass materials, and after a designated stress relaxation period, the strain is diminished to a lower level. Notably, an increase in the stress level is observed before it ultimately decreases. The authors scrutinize this phenomenon through the use of in-situ synchrotron X-ray experiments and relaxation kinetics. This comprehensive examination of the seemingly universal phenomenon is commendable as it spanned across different types of glasses, namely two metallic glasses and a polymer glass (PVC). This paper offers substantial insights and is well-structured. I'd recommend its publication post proper revisions on the following points:

Response: We thank the reviewer for his/her favorable assessment of our work, and for kindly providing thoughtful feedback and suggestions to improve the manuscript. The following is a point-by-point response to the specific comments.

English Language Refinement:

Question 1. Page 2: The sentence "One kernel is the complex relaxation behaviors derived from nonequilibrium thermodynamics" requires revision. The term 'kernel' seems incongruous in this context and there are grammatical inconsistencies.

Response: Thank you for your correction, and we have modified this sentence as: "*One crucial aspect is the complex relaxation behaviors that arise from nonequilibrium thermodynamics.*"

Question 2. Page 2: The phrase "For structural glasses, recent results suggest that the stress field is equal to the thermal effect for triggering the flow" lacks clarity. The implication of "equal" in this context is ambiguous.

Response: Thank you for your correction, and we have modified this sentence as: "*For structural glasses, recent results suggest that the stress and the thermal effects both contribute similarly to triggering flow.*"

Guan *et al.* have obtained a phase diagram of viscosity in the temperature-stress fields for amorphous alloy systems (*Phys. Rev. Lett.* 104, 205701 (2010). *Acta Physica Sinica* 66, 176112 (2017)), as depicted in Fig. R10. When the system state is characterized by viscosity, defining the critical viscosity at which the system undergoes the glass transition under stress, the phase space can be divided into regions of the glassy state and the liquid/flowing state. Through this phase diagram, it is clear that there are at least three ways to induce flow in amorphous systems: 1) keeping the stress constant and increasing the temperature, 2) keeping the temperature constant and increasing the stress, and 3) simultaneously changing the temperature and the stress. The analogous finds were proposed in other "glasses" including silicate and polymer glasses, granular materials, soils, and emulsifiers (*Nat. Mater.* 21, 404-409 (2022). *Nat. Mater.* 21, 404-409 (2022). *Nature* 411, 772-775 (2001)). These findings suggest that increasing stress and increasing temperature contribute similarly to triggering flow of glas.

Fig. R10. The counterplot of viscosity as a function of temperature and stress of metallic glasses.
Copy from Fig.3 of PRL 104, 205701 (2010).

Question 3. Page 2: The assertion that the Kovacs-like memory effect is valid for structural glasses could benefit from additional specificity concerning the type of materials where this memory effect might be observed.

Response: We did additional experiments in PTFE polymer glass, similar stress memory effect has been identified (see Fig. R11). Herein, we could infer that the Kovacs-like stress memory effect is applicable to various structural glasses, as supported by our observations in TiZrHfCuNiBe, $(\text{Fe}_{11}\text{Zr}_1)_{91.2}\text{B}_{8.8}$ metallic glasses, and PVC and PTFE polymer glasses. After conducting in-depth research, we have discovered that the Kovacs-like memory effect originates from the runback of anelasticity deformation. Since anelasticity is a universal characteristic for structural glasses (*Prog. Mater. Sci.* 106, 100561 (2019). *Phys. Rev. Lett.* 99, 135502 (2007). *Phys. Rev. B* 67, 092202 (2003)), it is reasonable to expect that memory effect will happen in all structural glasses.

Fig. R11. Stress memory effect for the Poly tetra fluoroethylene (PTFE) polymer glass at 303 K

with $\varepsilon_1 = 0.5\%$ and $\varepsilon_2 = 0.3\%$.

Question 4. Page 4: The authors should cite a reference for the Kohlrausch-Williams-Watts (KWW) expression.

Response: Thank you for your suggestion. We have cited the relevant references [3, 35, 36] for the KWW (Kohlrausch-Williams-Watts) expression in the modified version.

Question 5. Clarification of Memory Effect Understanding:

The understanding of the memory effect appears to hinge on the anelastic deformation process, colloquially known as beta relaxation. The X-ray experiment results, particularly the $G(r)$ and the evolution of the Full Width at Half Maximum (FWHM) of its first peak, suggest a correlation with the memory effect. However, the types of atomic displacements associated with the stress level rise are not clearly outlined.

Response: Thank you for your professional comments. We have performed calculations of interatomic bond lengths and weight factors, as presented in Supplementary Table 1. By combining these results with the peak positions of the two sub-peaks within the first peak in the pair distribution function ($G(r)$), we have conducted a detailed analysis of the types of atomic displacements. This analysis is highlighted in red in the modified version of the manuscript, and it is also presented in the subsequent section:

“According to the interatomic bond lengths and weight factors provided in Supplementary Table 1, the left sub-peak at approximately 2.70 Å can be primarily attributed to the Hf-Cu and Hf-Ni atomic pairs, while the right sub-peak at around 3.10 Å is mainly dominated by the Zr-Hf, Hf-Hf, and Ti-Hf atomic pairs.”

Supplementary Table 1. The possible nearest-neighbor pairs in $\text{Ti}_{16.7}\text{Zr}_{16.7}\text{Hf}_{16.7}\text{Cu}_{16.7}\text{Ni}_{16.7}\text{Be}_{16.7}$ metallic glass and their theoretical inter-atomic bond lengths $R_{ij}^0(\text{Å})$. Atomic radii (Ti-1.47 Å, Zr-1.60 Å, Hf-1.59 Å, Cu-1.28 Å, Ni-1.25 Å and Be-1.13 Å). w_{ij} denotes the weight factors of the atomic pairs calculated at $Q = 0\text{Å}^{-1}$.

$i-j$	$R_{ij}^0(\text{Å})$	w_{ij}
Zr-Hf	3.19	0.151
Hf-Hf	3.18	0.134
Hf-Cu	2.87	0.111
Hf-Ni	2.84	0.107
Ti-Hf	3.06	0.081
Zr-Cu	2.88	0.063
Zr-Ni	2.85	0.060
Ti-Zr	3.07	0.046
Cu-Ni	2.53	0.044

Zr-Zr	3.2	0.043
Ti-Cu	2.75	0.034
Ti-Ni	2.72	0.032
Cu-Cu	2.56	0.023
Ni-Ni	2.5	0.021
Hf-Be	2.72	0.014
Ti-Ti	2.94	0.012
Zr-Be	2.73	0.008
Cu-Be	2.41	0.006
Ni-Be	2.38	0.005
Ti-Be	2.6	0.004
Be-Be	2.26	0.000

Question 6. Distinction of Anelastic and Visco-plastic Deformation:

There is an apparent lack of clarity on how the authors differentiated between anelastic and viscoplastic deformation in the three different types of glasses during the relaxation process in this study.

Response: Thank you for the comments. We have provided additional clarity to differentiate between anelastic and viscoplastic deformation based on the strain recovery curves observed upon stress removal, as highlighted in red color in Supplementary Fig. 7.

“Supplementary Fig. 7 displays the strain recovery curves upon stress removal (switching from $\sigma > 0$ to $\sigma = 0$), highlighting the presence of three strain components: pure elastic strain (ϵ_{el}), anelastic strain (ϵ_{an}), and viscoplastic strain (ϵ_{vp}). Anelastic deformation (ϵ_{an}) refers to the time-dependent and recoverable strain observed in a material after the removal of applied stress. In contrast, viscoplastic deformation (ϵ_{vp}) signifies permanent deformation that cannot be recovered upon stress removal.”

Fig. R12. Schematic of the three deformation components observed in recovery experiments without applied force after stress relaxation below T_g (ϵ_{el} : elastic, ϵ_{an} : anelastic, ϵ_{vp} : viscoplastic).

Copy from the Supplementary Fig. 7(a).

Question 7. Link to Theoretical Discussion:

In the introduction, the authors refer to theories addressing the relaxation process in glasses and the memory effect (for instance, ref. 15). Nevertheless, in the discussion section, it remains unclear whether the obtained data supports any specific theoretical perspectives of the memory effect or if these results are extraneous to the discussions of the Kovacs memory effect. The authors assert near the conclusion that both memory effects originate from the same structural source, but it is unclear if they can be described within the same theoretical framework.

Response: Thank you for your professional comments. We have incorporated the discussion on the connection between our data and the relevant theoretical perspectives of the memory effect. The corresponding sections have been highlighted in red in the revised version.

The additional discussion on the theoretical aspect: “This aligns with the thriving of the Kovacs memory effect in a deep aging state characterized by a large activation entropy S^* . The herein reversible β events also support the rational presence of the reversible faster relaxing element in the phenomenological Amir-Oreg-Imry (AOI) model ^{1,2}, which has been widely used to explain the nonmonotonic relaxation phenomenon ^{2, 22, 23, 54}.”

The detailed discussion is as follows:

First, the Kovacs memory effect has been found to occur in a deep aging state characterized by a large activation entropy S^* , as denoted by filled circles with the curved arrow in Fig. R13a (taken from the “for instance, ref. 15”. *Phys. Rev. Lett.* 125, 135501 (2020)). Referring to the corresponding diagram of relaxation modes (Fig. R13b), the deep aging state denotes the collective motion process towards the α mode.

Based on our obtained data, the Kovacs-like stress memory effect is observed to be prominent in the high collective motion process validated by larger activation energy and micro-plastic deformation. Our results do not contradict the previous theoretical perspectives (“for instance, ref. 15”). Furthermore, we also find that the essence of the memory effect is the runback of reversible anelastic events, which provided a clearer physical image of the memory effect.

Fig. R13. a Schematic map of enthalpy changes during glass annealing in $\text{Au}_{49}\text{Cu}_{26.9}\text{Ag}_{5.5}\text{Pd}_{2.3}\text{Si}_{16.3}$ metallic glass (*Phys. Rev. Lett.* 125, 135501 (2020)). As the enthalpy decreases during isothermal annealing at low temperatures, the as-cooled glass first experiences a small S^* stage and then transitions into a large S^* stage. When the annealed glass jumps to a higher temperature, the

enthalpy continues to decrease if the jump reaches a relaxation stage with a small S^* , but there is a memory effect if the jump reaches a relaxation stage with large S^* . **b** the temperature-enthalpy diagram of different relaxation modes of $\text{Au}_{49}\text{Cu}_{26.9}\text{Ag}_{5.5}\text{Pd}_{2.3}\text{Si}_{16.3}$ metallic glass²⁹ (*Proc. Natl. Acad. Sci. U.S.A.* 120, e2302776120 (2023)).

Second, the Amir-Oreg-Imry (AOI) phenomenological model has been widely employed to explain the nonmonotonic relaxation/Kovacs-like memory (*Phys. Rev. Lett.* 118, 085501 (2017). *Phys. Rev. Lett.* 124, 168002 (2020)) effect observed in other complex disordered systems. Within the AOI framework, the relaxation dynamics of glassy systems are facilitated by a spectrum of uncoupled modes that exhibit exponential relaxation ($P(\lambda) \sim 1/\lambda$), with the density of these modes being inversely proportional to their relaxation timescales ranging from $1/\lambda_{\max}$ to $1/\lambda_{\min}$, as depicted in Fig. R14. In this state, the system demonstrates dual characteristics: it retains the memory of the initial unstrained state through the slow relaxation elements, while also adapting to the strained state ϵ and **being capable of relaxation in opposite directions through the fast relaxation elements** (*Phys. Rev. Lett.* 118, 085501 (2017). *Phys. Rev. Lett.* 124, 168002 (2020)). It is important to note that this is an assumption/inference in the phenomenological model. However, our experimental evidence, including strain recovery and *in-situ* high-energy X-ray diffraction, supports the notion that the mechanical memory effect arises from the runback of the reversible anelastic components/ β event.

Fig. R14. The different relaxation times framework with the Amir-Oreg-Imry (AOI) distribution of relaxation rates. *Copy from the Fig. 2 of Phys. Rev. Lett.* 124, 168002 (2020).

Thirdly, both the temperature-stimulated and stress-stimulated memory effects can be described using the same theoretical framework: “The classical Kovacs memory effect has a typical three-temperature protocol^{5, 6, 19, 25, 34, 55}, say T_0 , T_2 and T_1 , where T_0 is the initial equilibrated temperature, T_1 and T_2 are annealing temperatures. The Kovacs memory effect happens only as

$T_0 > T_2 > T_1$, while herein the stress memory effect happens only $\varepsilon_0 < \varepsilon_2 < \varepsilon_1$. Inspired by the theoretical works of Prados *et al.*^{14, 19, 26}, the contribution of the temperature perturbation to the glass system is alike to the stress/strain by considering $\beta E \propto \beta \sigma \varepsilon$. Here $\beta = \frac{1}{k_B T}$, and k_B is the Boltzmann constant. E represents the energy, which is basically proportional to $\sigma \varepsilon$ ¹⁹. Then, we can understand the effect of applied strain ε on the glass system as a perturbation β in the Kovacs memory effect. For instance, the *low-to-high* temperature annealing leads to a decrease in β , which is equal to the stimulation of *high-to-low* strain annealing in the protocol under a fixed temperature (Fig. 1a). The convergence of the two memory effects stems from their common structural origin, as the classical Kovacs memory effect also may arise from the recovery of reversible loosely packed regions^{56, 57}.”

Overall, we have carefully addressed all the suggestions and comments from the reviewers in the revised manuscript. Relevant discussions have been added to make the work more comprehensive to attract broader interests from readers. Hope it is ready for publication. Thank you very much!

With all my best wishes,

Sincerely yours,

Junqiang Wang

Reviewers' Comments:

Reviewer #1:

Remarks to the Author:

In the revised version of the paper “Strain-driven Kovacs-like memory effect in glasses”, the authors have addressed my concerns about the physical interpretation of their results, in view of previous existing work (mainly theoretical) in the literature. Now the results are correctly interpreted, and a complete analogy between the jumps in the temperature of the classic Kovacs’s protocol and the jumps in the strain considered by the authors.

The authors have also clarified my other main concern, the use of a protocol in which the value of the monitored quantity after the second jump does not correspond to equilibrium (as shown in cases B’, D’, and E’, of Fig. R3 in the authors’ reply letter). It is true, as commented by the authors, that this more generic protocol was also used by Kovacs in some of his seminal works, although currently the term “Kovacs memory effect” is mainly employed for the protocol corresponding to case B’ in the same figure of the authors’ reply.

As a minor point, in the supplemental material, the authors present different fits of experimental relaxation data to the KWW or stretched exponential law. It is curious that the authors’ findings, with the exponent β close to $1/2$ and an Arrhenius behaviour of the relaxation time, are compatible with theoretical works in which the KWW behaviour is found in simple models—like Ising models. See for instance JCP 63, 5445 (1975); Commun Math Phys 125, 3 (1989); Physica A 197, 569 (1993); PRE 53, 458 (1996). Therein, the emergence of an exponent $\beta \approx 1/2$ can be often understood in terms of diffusion of defects; commenting on this point could be interesting for the potential readers of the paper.

After the authors’ revision to address my—and also the other referees’—comments, the relevance of the results is clearer and the quality of the presentation has been substantially improved. As a consequence, I believe that the revised manuscript may belong in Nature Communications.

Reviewer #2:

Remarks to the Author:

The authors have addressed my concerns.

I am happy to recommend this paper for publication.

Reviewer #3:

Remarks to the Author:

In their revised submission, Tong et al. have significantly revamped their manuscript. The authors have undertaken extensive additional research, supplemented the paper with new figures, and meticulously addressed each concern raised by the reviewers. I am especially impressed with their comprehensive response to my previous comment concerning the absence of ties to contemporary theoretical discussions. Given the thoroughness and quality of their revisions, I wholeheartedly recommend the publication of this work in its current form.

Dear Editors and Reviewers:

Thank you for your patience and valuable contribution in reviewing our manuscript (Manuscript ID: NCOMMS-23-17820A). We appreciate the reviewers' approval and the editor's decision to publish our work in *Nature Communications*.

Point-by-point Responses:

Part I: for Reviewer 1

Reviewer #1 (Remarks to the Author):

In the revised version of the paper “Strain-driven Kovacs-like memory effect in glasses”, the authors have addressed my concerns about the physical interpretation of their results, in view of previous existing work (mainly theoretical) in the literature. Now the results are correctly interpreted, and a complete analogy between the jumps in the temperature of the classic Kovacs’s protocol and the jumps in the strain considered by the authors.

The authors have also clarified my other main concern, the use of a protocol in which the value of the monitored quantity after the second jump does not correspond to equilibrium (as shown in cases B’, D’, and E’, of Fig. R3 in the authors’ reply letter). It is true, as commented by the authors, that this more generic protocol was also used by Kovacs in some of his seminal works, although currently the term “Kovacs memory effect” is mainly employed for the protocol corresponding to case B’ in the same figure of the authors’ reply.

As a minor point, in the supplemental material, the authors present different fits of experimental relaxation data to the KWW or stretched exponential law. It is curious that the authors’ findings, with the exponent β close to 1/2 and an Arrhenius behaviour of the relaxation time, are compatible with theoretical works in which the KWW behaviour is found in simple models—like Ising models. See for instance JCP 63, 5445 (1975); Commun Math Phys 125, 3 (1989); Physica A 197, 569 (1993); PRE 53, 458 (1996). Therein, the emergence of an exponent $\beta \approx 1/2$ can be often understood in terms of diffusion of defects; commenting on this point could be interesting for the potential readers of the paper.

After the authors’ revision to address my—and also the other referees’— comments, the relevance of the results is clearer and the quality of the presentation has been substantially improved. As a consequence, I believe that the revised manuscript may belong in *Nature Communications*.

Response: We appreciate the positive assessment of our work and your recommendation for publication.

Regarding the minor point in the supplemental material about the exponent β_{kww} approaching 1/2, we have added a discussion in the Supplemental material as follows: “During the initial relaxation stage, β_{kww} is close to 0.5, a phenomenon often associated with the diffusion of defects.”. Upon conducting in-depth research (Fig. 4 in the main text), we have confirmed that the initial relaxation stage corresponds to the β relaxation process, which has been validated as the self-diffusion of the smallest constituting atoms in metallic glass (Physical Review Letters, 2012, 109(9): 095508). That coincides well with your comment. Lastly, thank you for providing thoughtful feedback and suggestions to improve the manuscript.

Part I: for Reviewer 2

The authors have addressed my concerns.

I am happy to recommend this paper for publication.

Response: Thank you very much for your approval. We sincerely appreciate the reviewer’s comments and suggestions to improve the significance of our work.

Part I: for Reviewer 3

In their revised submission, Tong et al. have significantly revamped their manuscript. The authors have undertaken extensive additional research, supplemented the paper with new figures, and meticulously addressed each concern raised by the reviewers. I am especially impressed with their comprehensive response to my previous comment concerning the absence of ties to contemporary theoretical discussions. Given the thoroughness and quality of their revisions, I wholeheartedly recommend the publication of this work in its current form.

Response: We sincerely appreciate the reviewer’s endorsement. All the comments and suggestions are valuable to raise the quality and novelty of our work.